# BREADCRUMBS REASONING: MEMORY-EFFICIENT REASONING WITH COMPRESSION BEACONS

## ABSTRACT

The scalability of large language models for long-context reasoning is severely constrained by the linear growth of their Transformer key-value cache, which incurs significant memory and computational costs. We posit that as a model generates reasoning tokens, the informational value of past generated tokens diminishes, creating an opportunity for compression. In this work, we propose to periodically compress the generation KV cache with a learned, special-purpose token and evict compressed entries. We train the model to perform this compression via a modified joint distillation and reinforcement learning (RL) framework. Our training method minimizes overhead over the conventional RL process, as it leverages RL outputs for distillation. Empirically, our method achieves a superior memory-accuracy Pareto frontier compared to both the model without cache compression and training-free compression techniques.

## 1 INTRODUCTION

Reasoning through token generation allows large language models (LLMs) to solve arbitrarily complex problems with a fixed depth architecture (Merrill & Sabharwal, 2023), by scaling the compute invested through the generation of more tokens (i.e., test-time scaling) (Snell et al., 2024). This practice carries high computational costs, because of the self-attention design of Transformers (Bahdanau et al., 2014; Vaswani et al., 2017; Keles et al., 2023). Not only it relies on simply generating many more tokens, but later tokens require computation over the representations of all previous tokens, incurring higher time complexity and necessitating increasing memory costs.

However, not all past representations are equally important. For example, the details of a previously explored attempt at a solution are likely not critical, as long as the model retains some signal that

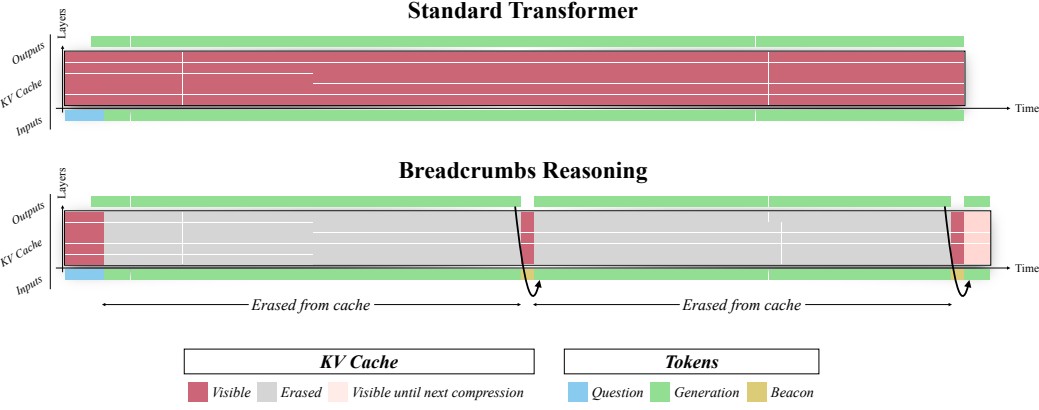

**Figure 1: Breadcrumbs Reasoning,** with a compression ratio $c = 16$. To save memory during inference, a window of $c$ tokens is periodically compressed into a single beacon token. The original KV cache entries for the window are then evicted, leaving only a compact 'breadcrumb' that summarizes the preceding reasoning steps.

advises it to avoid exploring this same failed path again. We propose to jointly learn to reason, compress, and discard previously computed representations along reasoning chains.

The key to our approach is to substitute previously computed key-value (KV) cached representations with significantly more compact representations, inspired by how activation beacons are used for long-context compression (Zhang et al., 2025). We train these representations to contain the information from past tokens that is necessary for continuing the reasoning process to solve the task the model is given, allowing us to evict from the KV cache most previously computed representations. The challenge is to combine the training of these beacons into the reinforcement learning (RL) (Sutton & Barto, 2018) process that makes length-based reasoning possible, a fundamentally different process than the pre-training that enables conventional long-form generation. We design a joint RL-distillation approach, where we train the original non-compression policy using the conventional RL process with a verifier for reward computation, and concurrently distill it into a policy that jointly compresses and reasons.

We evaluate our method, Breadcrumbs Reasoning (Figure 1), on the Qwen2.5-1.5B and Phi-4-Mini models across three challenging reasoning benchmarks: Countdown (Gandhi et al., 2024), LinSys, and StarGraph (Bachmann & Nagarajan, 2024). We compare against a strong, uncompressed teacher policy trained with RL, as well as four training-free cache eviction baselines: PyramidKV (Cai et al., 2025), SnapKV (Li et al., 2024), TOVA (Oren et al., 2024), and StreamingLLM (Xiao et al., 2023). Our experiments reveal several key findings. Demonstrating a clear Pareto improvement, Breadcrumbs Reasoning enables effective test-time scaling by generating longer reasoning chains to match or exceed the teacher's accuracy within a fixed memory budget, while still retaining 65.1–89.8% of the original performance when using 2–32x less memory at a fixed generation length. Notably, at inference time, it is possible to choose the compression ratio for each given input based on the memory budget and required performance. In contrast, the training-free baselines consistently underperform, stressing the necessity of a learned compression scheme for complex reasoning. We also validate our training strategy, showing that our joint RL-distillation approach matches or outperforms a more complex two-stage training pipeline, confirming its efficiency. Our code will be released upon publication.

## 2 RELATED WORK AND BACKGROUND

Generation in Transformers-based LLMs requires reasoning over and storing a key-value (KV) cache. This entails high memory (i.e., space) and time costs, which increase as the context (i.e., the number of previous tokens) increases. Therefore long-form generation costs suffer not only from the fundamental need to generate more tokens via more steps, but also from the increasing cost of each such step. KV compression is a solution avenue that is receiving significant research attention (LI et al., 2025).

An important thread within this compression literature is training models to perform KV cache compression. Nawrot et al. (2024) train models to compute importance scores, which are then used to store averaged KV cache entries instead of the original entries. Other methods train the Transformer-based LLMs themselves to summarize past KV entries (Mu et al., 2023; Chevalier et al., 2023; Zhang et al., 2025), so more compact representations can be retained. Our approach is inspired by the activation beacons method (Zhang et al., 2025), but with significant simplifications and adaptation for reasoning. We do away with the chunk and sub-chunk distinction, and eliminate the addition of specialized attention mechanisms. Rather, we adopt a flat segmentation into blocks, use the standard Transformer attention mechanism, and remove KV cache entries every time a beacon is processed (e.g., ex-ante). These modifications do not only simplify the implementation, but also allow for immediate eviction of cache entries, instead of a delayed one. More broadly, a drawback of these learned methods is that they require fine-tuning on a considerable amount of general-purpose pre-training data. We design a joint reasoning-compression training approach, which adds minimal overhead over the existing reasoning training processes.

An alternative to training-based methods are training-free methods that perform compression at generation time. They can be divided into two main categories. The first is that of sliding-window approaches, which limit the KV cache by only including a sliding window plus an additional subset of tokens. Particularly simple and effective is StreamingLLM (Xiao et al., 2023), which finds that including a few initial *sink* tokens in addition to the window recovers most of the uncompressed

---

**Algorithm 1** Breadcrumbs Reasoning

---

**Input:** Transformer-based policy $\pi_{\mathrm{BR}}$, beacon token $b$, prompt tokens $\bar{q}$, stop token $s$, compression ratio $c$. Let $\mathrm{KV}_{\pi_{\mathrm{BR}}}$ be the persistent KV cache of the policy model $\pi_{\mathrm{BR}}$.
**Output:** $\bar{x}$
1: **Initialize:** Encode $\bar{q}$ through $\pi$
2: **for** $i = 0, 1, 2, \ldots$ **do**
3:      $x_i \sim \pi_{\mathrm{BR}}(\cdot | \bar{q}, \bar{x})$          ▷ *Sample the next token.* $\mathrm{KV}_{\pi_{\mathrm{BR}}}$ *is updated internally.*
4:      $\bar{x} \leftarrow \bar{x} + x_i$          ▷ *Concatenate the sampled token to the end of the output.*
5:      **if** $x_i = s$ **then**          ▷ *Check for the generation stopping token.*
6:          **break**
7:      **if** $i > 0$ **and** $i \bmod c = 0$ **then**
8:          Encode $b$ through $\pi_{\mathrm{BR}}(\cdot | \bar{q}, \bar{x})$     ▷ *Updates* $\mathrm{KV}_{\pi_{\mathrm{BR}}}$ *with the entry for the compression token $b$.*
9:          $\mathrm{KV}_{\pi_{\mathrm{BR}}} \leftarrow \mathrm{KV}_{\pi_{\mathrm{BR}}}[: -c - 1]$    ▷ *Drop the KV cache entries of the most recent $c$ tokens.*
10:            $+\mathrm{KV}_{\pi_{\mathrm{BR}}}[-1]$          ▷ *But, keep the entry of the beacon $b$.*
11: **return** $\bar{x}$

---

performance. The second type does not constrain window tokens to remain in the cache, but rather tries to select the empirically more important for attention computations, as in H2O (Zhang et al., 2023) or TOVA (Oren et al., 2024). Other approaches that can be considered part of this category focus on prefill compression, reducing tokens by using the attention scores of the window of most recent token of the entire prefilling prompt (Cai et al., 2025; Li et al., 2024). While these latter methods are designed for prefill and do not compress generation tokens directly, it could be possible to apply them repeatedly to compress at generation time.

An orthogonal approach to reducing the KV cache size is reducing Chain-of-Thought reasoning length (Aggarwal & Welleck, 2025; Kang et al., 2025; Ma et al., 2025; Shen et al., 2025; Yan et al., 2025; Munkhbat et al., 2025; Xia et al., 2025). These methods primarily aim to directly shorten reasoning traces. This is distinct from our objective of dynamic KV cache compression, which focuses on extracting critical information to manage cache sizes effectively during the reasoning process itself. KV cache compression methods, like ours, could be applied in combination with those methods to achieve even higher levels of efficiency.

## 3 METHODOLOGY

Breadcrumbs Reasoning (BR) periodically computes compressed representations of KV cache entries and evicts them from the KV cache. We design a training process that adds relatively little overhead on top of the conventional reasoning RL process. The learned policy model effectively reasons through token generation and concurrently compresses KV cache representations.

### 3.1 BREADCRUMBS REASONING

Our compression scheme uses the Transformer architecture itself for both compression and reasoning. We generate tokens following the same procedure as a vanilla Transformer-based language model, but periodically compute compressed representations of past KV cache entries, and evict these entries from the cache. Algorithm 1 describes the process. We add a special token $b$ to the model vocabulary and embedding matrix, to mark when the model should compute compressed representations of past tokens. We input this token every $c$ tokens, where $c$ is the target compression ratio. The KV cache entries for the token $b$ form the compressed representation, and we drop the entries for previous $c$ tokens. Immediately after the beacon $b$, we force the next input token to be the last sampled token before $b$ was given as input. This is equal to continuing conventional generation. Figure 1 visualizes this process to illustrate the space savings. Roughly speaking, this process leaves a trace of "reasoning breadcrumbs" behind, instead of long, detailed, and eventually irrelevant information.

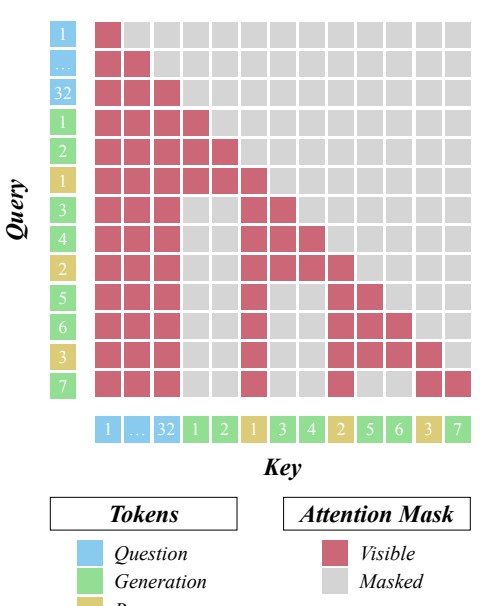

Figure 2: **Attention mask used to enforce compression during training.** Each token after the question can attend to the initial question tokens, all previous beacons, and earlier generation tokens within the same window, i.e., since the most recent preceding beacon. This encourages the model to compress relevant past context into the beacons to support future generations.

## 3.2 JOINT RL-DISTILLATION TRAINING

Typically, LLMs are trained to solve reasoning tasks through RL (Shao et al., 2024; DeepSeek-AI, 2025; Lambert et al., 2025). Applying this process as is to a breadcrumbs reasoning policy is technically possible, but is unlikely to lead to effective learning. This is because, before training, the model does not have compression ability, so incorporating the compression token and cache eviction will significantly damage its functionality, and it will observe no positive reward during RL.

We use a surrogate teacher policy $\pi_{\text{RL}}$ that does not compress and is trained through RL to perform the reasoning task, and distill it into the breadcrumbs policy $\pi_{\text{BR}}$. By learning to imitate the surrogate policy $\pi_{\text{RL}}$, our target breadcrumbs policy $\pi_{\text{BR}}$ simultaneously learns to compress and to perform the new reasoning task. This process relies on the trajectories sampled during RL, so no expensive sampling of trajectories beyond the conventional RL process is needed. This procedure minimizes the overhead over a standard two-steps procedure, where either (a) $\pi_{\text{RL}}$ is completely trained before generating new data for $\pi_{\text{BR}}$ to learn to compress, or (b) $\pi_{\text{BR}}$ is trained to compress through extensive general data and only after that learns the new reasoning task.

Given a token trajectory $\bar{x}$ sampled from $\pi_{\text{RL}}$, the distillation loss is defined as the average token-level KL divergence between the uncompressed policy $\pi_{\text{RL}}$ and the compressed policy $\pi_{\text{BR}}$:

$$L(\bar{x}) = \frac{1}{|\bar{x}|} \sum_{i=1}^{|\bar{x}|} D_{KL}\left(\pi_{\text{RL}}(x_i|\bar{x}_{<i}) \parallel \pi_{\text{BR}}(x_i|\bar{x}_{<i})\right) \tag{1}$$

$$= \frac{1}{|\bar{x}|} \sum_{i=1}^{|\bar{x}|} \sum_{k=1}^{|V|} \pi_{\text{RL}}(x_i = k|\bar{x}_{<i}) \log \frac{\pi_{\text{RL}}(x_i = k|\bar{x}_{<i})}{\pi_{\text{BR}}(x_i = k|\bar{x}_{<i})}$$

The total loss is then computed as the average over a batch of $B$ trajectories:

$$L = \mathbb{E}_{\bar{x} \sim \pi_{\text{RL}}}\left[L(\bar{x})\right] \approx \frac{1}{B} \sum_{b=1}^{B} L(\bar{x}_b)$$

When training $\pi_{\text{BR}}$, we only compute the gradient of $L$ with respect to the parameters of $\pi_{\text{BR}}$. The parameters of the teacher policy $\pi_{\text{RL}}$ are kept frozen during this update step, as $\pi_{\text{RL}}$ should not adapt to $\pi_{\text{BR}}$.

For parallel and efficient training, at training time $\pi_{\text{BR}}$ does not execute the KV cache removal strategy (Section 3.1), but simulates it by masking compressed tokens. Figure 2 visualizes the attention mask that results from the training masking pattern. Therefore, $\pi_{\text{BR}}$ learns to use the

beacon token $b$ and to compress by aligning its next token distribution to the next token distribution of a policy without compression, $\pi_{\text{RL}}$. Thanks to the attention mask, the model at training time needs to learn how to retain useful information through the beacons' activations, as future tokens could not otherwise use it.

# 4 EXPERIMENTAL SETUP

**Tasks**   We use three reasoning tasks that the initial models solve with only a very low success rate:

*Countdown* (Gandhi et al., 2024): the task input is a tuple of numbers, and the goal is to create a sequence of arithmetic operations using a subset of the numbers to equal a target number. While this task requires the model to avoid repeating previous attempts, or it can enter an endless loop, the individual guesses are largely independent of one another.

*LinSys*: each problem consists of a linear equation system with a unique integer solution. The coefficients and variables are randomly generated. In contrast to Countdown, which emphasizes educated trial and error, this task more closely reflects structured reasoning: solving for one variable often enables solving for the next, resulting in a multi-step deductive process.

*StarGraph* (Bachmann & Nagarajan, 2024): the model is given a list of directed edges of a star graph (i.e., a graph with multiple branches of a fixed length all expanding from a central node) and a target end node. The model must output the full sequence of edges to get to the target node from the central node. This is a task auto-regressive models such as Transformers naturally struggle with, as observed by Bachmann & Nagarajan (2024) and Hu et al. (2025).

**Model and Training Details**   We utilize two models for our experiments: Qwen2.5-1.5B-Instruct (Team, 2025) and Phi-4-mini-instruct (Microsoft, 2025). In BR, we add to an additional token, the beacon $b$, to the vocabulary of these models and initialize its embeddings with the average of all other embeddings. For Qwen2.5-1.5B-Instruct, we train for 1000 steps for all tasks, while for Phi-4-mini-instruct, we train for 200 steps for StarGraph and LinSys and 1000 steps for Countdown. We use a batch size of 256 for both models. We use PPO (Schulman et al., 2017) as the RL algorithm for $\pi_{RL}$. We experiment with compression ratios $c$ of 2, 4, 8, 16, and 32 for breadcrumbs reasoning. We instantiate our training process in two different ways. In SR BR (Single Ratio Breadcrumbs Reasoning), each model is trained on a single compression ratio. In MR BR (Multi Ratio Breadcrumbs Reasoning), we train a single model on all compression ratios, by repeating each batch for all compression ratios before each model weights update. Then, the same model can be used at generation time with any desired ratio. The reward for an incorrect or not present answer is 0.0, a correct format of the final answer but an incorrect value gets a 0.1 reward, and a correct response gets a 1.0 reward.

**Baselines**   We compare our approach against two primary training-free baselines applied to $\pi_{\text{RL}}$: PyramidKV (Cai et al., 2025), SnapKV (Li et al., 2024), TOVA (Oren et al., 2024), and StreamingLLM (Xiao et al., 2023). Similar to our joint training setup, they also do not require more data than what $\pi_{\text{RL}}$ was trained on, and they also delete entries from the KV cache.[1] For a fair comparison, we adapt these methods to use an increasingly large KV cache size during generation, matching the memory footprint of our approach. This is to avoid potentially penalizing them with a smaller static cache size. In particular, given a compression ratio $c$, we allow the cache to grow by 1 every $c$ generated tokens. We set the sliding window of StreamingLLM and the observation window of SnapKV and PyramidKV to $c$ to match the budget given to our approach. We also set the sink size of StreamingLLM to the entire question. For SnapKV, PyramidKV, and TOVA, we set the initial size of the KV cache to the number of question tokens plus $c$, so that the first compression would only happen after at least $c$ tokens, similar to our method. For all methods, we test the same $c$ as for our policies. We also study the effect of two-step training, by first training the RL policy $\pi_{\text{RL}}$ and only later generating data for distillation. In both cases, we use the same number of data samples and training steps (256 samples per step, for 1k steps).

---

[1]We omit comparing to methods such as QUEST (Tang et al., 2024), because they are not focused on compression, but rather smart management of the GPU memory, an orthogonal approach to compression.

| | **QWEN** | | | | | | | | | | | | | | | | | |
| | COUNTDOWN | | | | | | LINSYS | | | | | | STARGRAPH | | | | | |
| COMPR. RATIO | 1 | 2 | 4 | 8 | 16 | 32 | 1 | 2 | 4 | 8 | 16 | 32 | 1 | 2 | 4 | 8 | 16 | 32 |
| *NO COMPRESSION* | | | | | | | | | | | | | | | | | | |
| TEACHER | 0.598 | - | - | - | - | - | 0.918 | - | - | - | - | - | 0.902 | - | - | - | - | - |
| *BREADCRUMBS REASONING* | | | | | | | | | | | | | | | | | | |
| SR BR (OURS) | - | 0.605 | 0.613 | 0.613 | 0.574 | 0.535 | - | **0.730** | **0.656** | 0.410 | 0.367 | 0.297 | - | 0.957 | **0.969** | 0.973 | 0.957 | 0.957 |
| MR BR (OURS) | - | **0.613** | **0.617** | **0.629** | **0.605** | **0.582** | - | 0.711 | 0.629 | **0.473** | **0.414** | **0.328** | - | **0.965** | 0.961 | **0.980** | **0.961** | **0.965** |
| *TRAINING-FREE COMPRESSION* | | | | | | | | | | | | | | | | | | |
| PYRAMIDKV | - | 0.445 | 0.215 | 0.117 | 0.078 | 0.012 | - | 0.000 | 0.000 | 0.000 | 0.000 | 0.000 | - | 0.605 | 0.504 | 0.516 | 0.477 | 0.539 |
| SNAPKV | - | 0.449 | 0.219 | 0.141 | 0.102 | 0.082 | - | 0.004 | 0.000 | 0.000 | 0.000 | 0.000 | - | 0.660 | 0.523 | 0.508 | 0.465 | 0.500 |
| TOVA | - | 0.574 | 0.289 | 0.172 | 0.188 | 0.207 | - | 0.000 | 0.000 | 0.000 | 0.000 | 0.000 | - | 0.664 | 0.457 | 0.445 | 0.430 | 0.441 |
| STREAMINGLLM | - | 0.012 | 0.023 | 0.027 | 0.051 | 0.094 | - | 0.000 | 0.000 | 0.000 | 0.000 | 0.000 | - | 0.055 | 0.055 | 0.031 | 0.047 | 0.117 |
| | **PHI** | | | | | | | | | | | | | | | | | |
| | COUNTDOWN | | | | | | LINSYS | | | | | | STARGRAPH | | | | | |
| COMPR. RATIO | 1 | 2 | 4 | 8 | 16 | 32 | 1 | 2 | 4 | 8 | 16 | 32 | 1 | 2 | 4 | 8 | 16 | 32 |
| *NO COMPRESSION* | | | | | | | | | | | | | | | | | | |
| TEACHER | 0.633 | - | - | - | - | - | 0.898 | - | - | - | - | - | 0.848 | - | - | - | - | - |
| *BREADCRUMBS REASONING* | | | | | | | | | | | | | | | | | | |
| SR BR (OURS) | - | **0.609** | **0.625** | **0.613** | **0.625** | 0.613 | - | 0.652 | 0.539 | 0.363 | 0.219 | 0.195 | - | **0.812** | **0.832** | **0.836** | **0.812** | **0.816** |
| MR BR (OURS) | - | 0.586 | 0.594 | **0.613** | 0.586 | **0.625** | - | **0.668** | **0.582** | **0.461** | **0.297** | **0.270** | - | 0.809 | 0.805 | 0.812 | 0.758 | 0.766 |
| *TRAINING-FREE COMPRESSION* | | | | | | | | | | | | | | | | | | |
| PYRAMIDKV | - | 0.484 | 0.238 | 0.184 | 0.020 | 0.000 | - | 0.016 | 0.000 | 0.000 | 0.000 | 0.000 | - | 0.637 | 0.402 | 0.305 | 0.336 | 0.367 |
| SNAPKV | - | 0.469 | 0.230 | 0.152 | 0.062 | 0.012 | - | 0.008 | 0.000 | 0.000 | 0.000 | 0.000 | - | 0.652 | 0.465 | 0.344 | 0.340 | 0.312 |
| TOVA | - | 0.230 | 0.066 | 0.051 | 0.047 | 0.098 | - | 0.000 | 0.000 | 0.000 | 0.000 | 0.000 | - | 0.801 | 0.562 | 0.457 | 0.453 | 0.395 |
| STREAMINGLLM | - | 0.016 | 0.031 | 0.109 | 0.156 | 0.312 | - | 0.000 | 0.000 | 0.000 | 0.000 | 0.000 | - | 0.180 | 0.020 | 0.031 | 0.098 | 0.102 |

Table 1: **Model performance on long-context reasoning tasks for a fixed generation cache size**. We report accuracy given a maximum cache size of 1,000 entries. **Bold** indicates the best result; underlined indicates the second best.

**Evaluation**    We generate evaluation answers for 256 held-out test examples for each task. In all settings, we fix the maximum KV cache size to 1,000 entries (i.e., tokens), which is also the maximum response length permitted during training. Generation is interrupted if this limit is reached.

## 5    RESULTS

We compare the Breadcrumbs Reasoning policy to an uncompressed RL policy and to the baselines in two different configurations. We first compare results with a set maximum cache size (Table 1, Table 6, and Figure 6). We also compare with a fixed maximum number of generation tokens (Table 2), the same used during training of $\pi_{\text{RL}}$.

**Test-Time Scaling with Breadcrumbs Reasoning**    Table 1 shows accuracy performance with a fixed maximum cache budget of 1,000 steps, while we refer to Table 6 in Appendix E for the accuracy area under the curve (AUAC) with varying the maximum cache size up to 1,000. Figure 6 shows the curves. Across most settings, BR recovers most of the performance of the teacher at a much lower memory cache budget, both in the single ratio and multi ratio settings. Except for LinSys, which remains challenging, all compression ratios outperform the teacher for most of the budgets. This is because the compressed models are able to effectively accommodate more reasoning steps within the same cache budget, allowing for more aggressive test-time compute scaling. On Countdown, both BR models outperform even the teacher at maximum KV cache budget.

We measure the trade-off between accuracy and KV cache consumption by measuring the area under the accuracy-cache size curve (Table 6). This metric integrates accuracy over the full range of cache sizes, and thus captures a model's overall robustness to a variety of cache constraints. A larger AUAC indicates that a method maintains high accuracy across a wider range of cache budgets, rather than only performing well with high memory usage. We can observe that BR outperforms even the teacher policy by 3.6–92.8% for Qwen and 10.9—219.7% for Phi across the different ratios and tasks. For high compression ratios, the number of generated tokens is much higher than during training (e.g., 32,000 tokens vs 1,000 for 32x compression ratio). This is evidence that compression

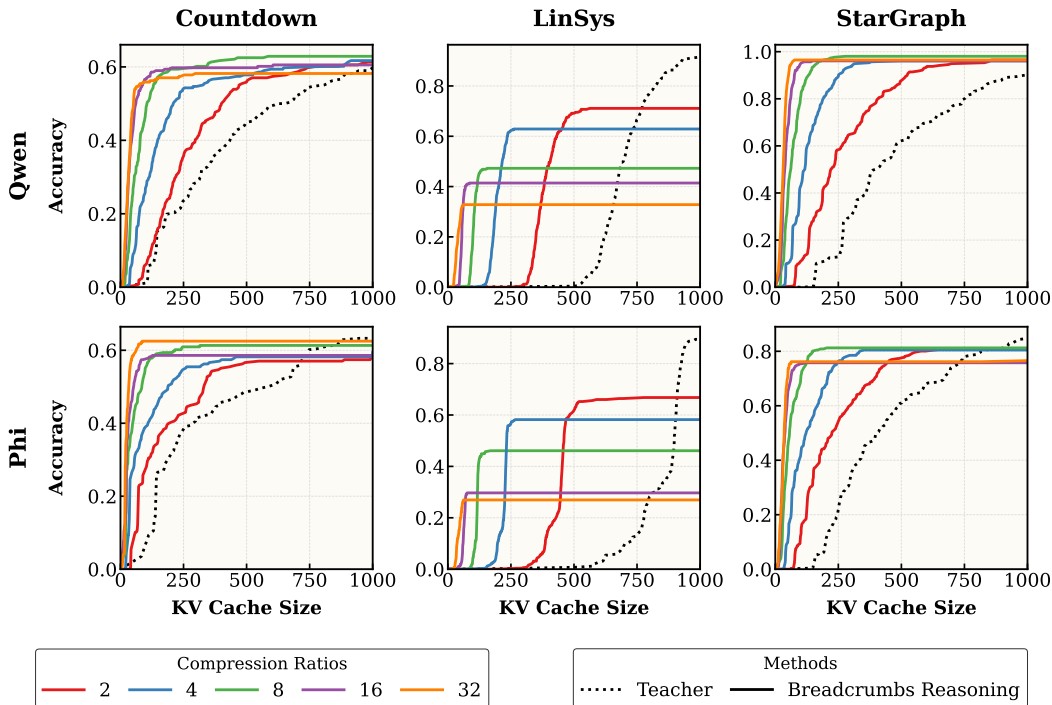

Figure 3: **Accuracy vs. KV Cache Size** Multi Ratio Breadcrumbs Reasoning retains most of the teacher's performance while using significantly fewer KV cache entries, even outperforming the teacher when setting a fixed maximum KV cache size in Countdown and StarGraph.

generalizes to much longer sequences than those observed during training, except for LinSys (and we refer to Figure 7 in the Appendix to visualize this effect). Crucially, despite potentially generating significantly more tokens (up to $32\times$), BR maintains an inference time comparable to the teacher and on average faster than all training-free baselines, as reported in Table 3.

Beyond matching teacher accuracy at full context, BR is far more efficient with cache usage. In Countdown and StarGraph, Figure 6 shows that BR nearly reaches or even exceeds the teacher's peak accuracy with fewer than 250 tokens in cache - less than a quarter of the teacher's required 1,000 tokens. For instance, in Phi–Countdown, BR surpasses 0.60 accuracy by cache size 100, while the teacher requires more than 800 tokens to reach the same level. A similar pattern holds in StarGraph, where BR consistently reaches >0.80 accuracy with only a few hundred tokens, whereas the teacher climbs much more gradually. The only clear exception is LinSys, where BR improves more slowly and is not able to completely close the gap to the teacher.

**Breadcrumbs Reasoning Retains Most of Uncompressed Performance**  We compare the BR policies across different compression ratios to the $\pi_{\mathrm{RL}}$ that we use as the distillation source. We maintain the same source policy for all our experiments to make this comparison valid. Table 2 reports the results for a fixed maximum generation length (i.e., time steps) of 1,000.

While on Countdown and StarGraph all compression ratios work well, performance on LinSys varies greatly across compression ratios. As expected, higher compression ratios lead to worse performance. For LinSys, BR lags behind the teacher for both models. We speculate this is due to differences in the reasoning challenge compared to Countdown and StarGraph. Overall, given a fixed response length, SR BR preserves performance across tasks by 67.1–94.0% for Qwen and 65.1–84.5% for Phi while using only 2–32x fewer KV cache entries at generation time.

**Multi-Ratio Training Robustness**  We analyze the performance of MR BR, where a single model is trained to handle all compression ratios simultaneously. One might expect that learning a policy capable of varying compression granularities would be a harder optimization problem, potentially

| | **QWEN** | | | | | | | | | | | | | | | | | |
|---|---|---|---|---|---|---|---|---|---|---|---|---|---|---|---|---|---|---|
| | COUNTDOWN | | | | | | LINSYS | | | | | | STARGRAPH | | | | | |
| COMPR. RATIO | 1 | 2 | 4 | 8 | 16 | 32 | 1 | 2 | 4 | 8 | 16 | 32 | 1 | 2 | 4 | 8 | 16 | 32 |
| NO COMPRESSION | | | | | | | | | | | | | | | | | | |
| TEACHER | 0.598 | - | - | - | - | - | 0.918 | - | - | - | - | - | 0.902 | - | - | - | - | - |
| BREADCRUMBS REASONING | | | | | | | | | | | | | | | | | | |
| SR BR (OURS) | - | **0.559** | **0.578** | 0.508 | 0.430 | 0.438 | - | **0.707** | **0.648** | 0.395 | 0.359 | 0.289 | - | **0.906** | 0.883 | 0.895 | 0.867 | 0.891 |
| MR BR (OURS) | - | **0.559** | 0.539 | **0.551** | **0.527** | **0.531** | - | 0.691 | 0.625 | **0.457** | **0.406** | **0.328** | - | 0.875 | **0.898** | **0.898** | **0.875** | **0.895** |
| TRAINING-FREE COMPRESSION | | | | | | | | | | | | | | | | | | |
| PYRAMIDKV | - | 0.438 | 0.211 | 0.113 | 0.078 | 0.012 | - | 0.000 | 0.000 | 0.000 | 0.000 | 0.000 | - | 0.598 | 0.504 | 0.516 | 0.477 | 0.539 |
| SNAPKV | - | 0.441 | 0.219 | 0.141 | 0.102 | 0.082 | - | 0.004 | 0.000 | 0.000 | 0.000 | 0.000 | - | 0.660 | 0.523 | 0.508 | 0.465 | 0.500 |
| TOVA | - | 0.535 | 0.289 | 0.172 | 0.188 | 0.207 | - | 0.000 | 0.000 | 0.000 | 0.000 | 0.000 | - | 0.648 | 0.457 | 0.441 | 0.430 | 0.441 |
| STREAMINGLLM | - | 0.008 | 0.012 | 0.012 | 0.051 | 0.094 | - | 0.000 | 0.000 | 0.000 | 0.000 | 0.000 | - | 0.043 | 0.016 | 0.012 | 0.012 | 0.047 |

| | **PHI** | | | | | | | | | | | | | | | | | |
|---|---|---|---|---|---|---|---|---|---|---|---|---|---|---|---|---|---|---|
| | COUNTDOWN | | | | | | LINSYS | | | | | | STARGRAPH | | | | | |
| COMPR. RATIO | 1 | 2 | 4 | 8 | 16 | 32 | 1 | 2 | 4 | 8 | 16 | 32 | 1 | 2 | 4 | 8 | 16 | 32 |
| NO COMPRESSION | | | | | | | | | | | | | | | | | | |
| TEACHER | 0.633 | - | - | - | - | - | 0.895 | - | - | - | - | - | 0.848 | - | - | - | - | - |
| BREADCRUMBS REASONING | | | | | | | | | | | | | | | | | | |
| SR BR (OURS) | - | **0.590** | **0.574** | 0.574 | **0.594** | 0.570 | - | **0.641** | 0.516 | 0.352 | 0.207 | 0.195 | - | **0.777** | **0.797** | **0.793** | **0.785** | **0.781** |
| MR BR (OURS) | - | 0.566 | 0.547 | **0.578** | 0.555 | **0.582** | - | 0.617 | **0.574** | **0.434** | **0.285** | **0.270** | - | 0.773 | 0.762 | 0.770 | 0.734 | 0.734 |
| TRAINING-FREE COMPRESSION | | | | | | | | | | | | | | | | | | |
| PYRAMIDKV | - | 0.477 | 0.234 | 0.184 | 0.020 | 0.000 | - | 0.016 | 0.000 | 0.000 | 0.000 | 0.000 | - | 0.613 | 0.402 | 0.305 | 0.336 | 0.367 |
| SNAPKV | - | 0.469 | 0.230 | 0.152 | 0.062 | 0.012 | - | 0.008 | 0.000 | 0.000 | 0.000 | 0.000 | - | 0.652 | 0.465 | 0.344 | 0.340 | 0.312 |
| TOVA | - | 0.219 | 0.066 | 0.051 | 0.047 | 0.098 | - | 0.000 | 0.000 | 0.000 | 0.000 | 0.000 | - | 0.773 | 0.559 | 0.453 | 0.453 | 0.395 |
| STREAMINGLLM | - | 0.016 | 0.012 | 0.039 | 0.137 | 0.297 | - | 0.000 | 0.000 | 0.000 | 0.000 | 0.000 | - | 0.156 | 0.016 | 0.031 | 0.078 | 0.102 |

Table 2: **Model accuracy on long-context reasoning tasks for a fixed generation length**. The metric shown is accuracy at a sequence length of $L = 1,000$. **Bold** indicates the best result; underlined indicates the second best.

leading to lower performance compared to the specialized SR BR models. However, our results indicate the opposite. MR BR not only retains the flexibility to operate at any ratio at inference time but also outperforms SR BR on average. For example, on the Qwen model in Table 1, MR BR achieves an average accuracy of 69.6% across all tasks and compression ratios, compared to 68.1% for SR BR. This suggests that the joint training facilitates information sharing, enabling the model to transfer effective compression strategies across varying ratios.

**Training-Free Methods Underperform** The training-free methods TOVA and StreamingLLM consistently underperform across tasks. On Countdown, their performance drops dramatically with higher compression (e.g., TOVA falls from 0.574 at 2× to 0.172 at 8× for Qwen; StreamingLLM remains below 0.32 across all settings). On StarGraph, the gap between the baselines and out approach is even more severe, with StreamingLLM dropping below 0.1 accuracy in nearly every configuration. In LinSys, both StreamingLLM and TOVA fail to reach even a single correct output, while SnapKV and PyramidKV are still below 2% accuracy even with the lowest compression ratio. These results highlight the limitation of training-free cache eviction methods: for tasks requiring long coherent reasoning chains, simply truncating past tokens eliminates essential intermediate steps, from which the model is unable to recover.

**Why Does BR Struggle with Linear Systems?** While BR retains most of the accuracy in Countdown and StarGraph, LinSys seems to plateau earlier, especially for larger compression ratios. The main difference between this task and the other two is that it requires careful arithmetic and algebraic manipulation. However, it could also be that beacons forget more in this task than in the others. We test the two possibilities (BR struggles with arithmetic, beacons fail to memorize) with an LLM-as-a-Judge (Zheng et al., 2023) pipeline (detailed in Appendix D) and report our findings in Figure 4. Interestingly, we find that the vast majority of mistakes are attributed to arithmetic mistakes, and their number increases significantly with increased compression. One potential factor is that compressing numbers might be harder, and also that our compression method might break some arithmetic circuits of the model, and that our training is not long enough to alleviate these issues.

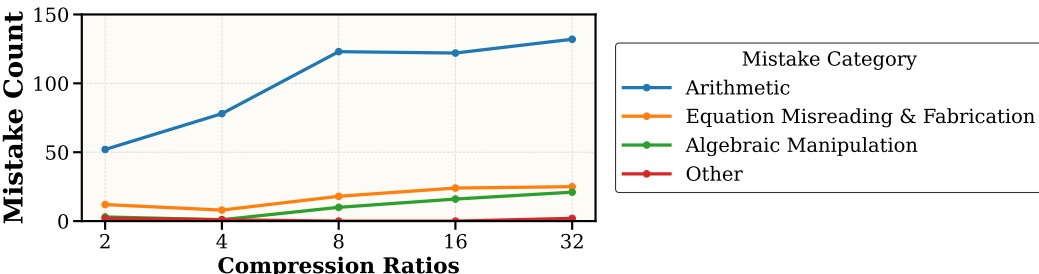

Figure 4: **Failure Mode Analysis on Linear Systems.** We classify error types of Qwen with SR BR across increasing compression ratios. The results indicate that the performance drop in LinSys is primarily driven by arithmetic errors, which scale significantly with compression, rather than a failure to memorize or retrieve information.

| | QWEN | | | | | PHI | | | | |
|---|---|---|---|---|---|---|---|---|---|---|
| METHOD | 2 | 4 | 8 | 16 | 32 | 2 | 4 | 8 | 16 | 32 |
| BREADCRUMBS REASONING | | | | | | | | | | |
| SR BR (OURS) | 1.990 | **1.546** | 1.593 | 1.646 | 1.468 | 2.029 | **1.486** | 1.518 | 1.506 | **1.164** |
| MR BR (OURS) | 1.999 | 1.665 | **1.507** | **1.396** | **1.366** | 1.821 | 1.574 | **1.307** | **1.306** | 1.220 |
| TRAINING-FREE COMPRESSION | | | | | | | | | | |
| PYRAMIDKV | **1.844** | 2.212 | 1.858 | 1.495 | 1.649 | 1.823 | 4.817 | 10.341 | 7.669 | 10.687 |
| SNAPKV | 2.607 | 2.364 | 1.748 | 1.664 | 1.504 | 1.738 | 4.347 | 14.438 | 10.575 | 15.234 |
| TOVA | 1.880 | 2.353 | 2.751 | 2.563 | 2.470 | **1.655** | 2.184 | 3.188 | 3.850 | 3.890 |
| STREAMINGLLM | 2.326 | 3.210 | 4.988 | 5.281 | 6.349 | 2.191 | 3.815 | 7.354 | 3.485 | 3.857 |

Table 3: **Latency increase across tasks.** The values represent the relative slowdown compared to the Teacher model (lower is better), given a maximum generation KV cache size of 1,000 entries (i.e., up to 32x tokens more with a ratio of 32). Columns represent the compression ratio. **Bold** indicates the best result; underlined indicates the second best.

**Joint RL-Distillation Training Matches or Outperforms a Two-Steps Training**    We compare two strategies for distilling from $\pi_{RL}$ in Figure 5 (for SR BR). In our joint RL-distillation training, BR is distilled online from the rollouts of the teacher $\pi_{RL}$ as it learns; in late distillation, compression policies are trained only on trajectories sampled from a final checkpoint of $\pi_{RL}$. BR achieves equivalent or superior performance in 26 of the 30 configurations tested, and very close performance on the other four. This demonstrates that it effectively piggybacks on the same data used to train $\pi_{RL}$. This eliminates the need for additional distillation data, minimizes training overhead, and does not impose the need to decide a priori a number of training steps for $\pi_{RL}$. We hypothesize that the superior performance may derive from the distributional shift of $\pi_{RL}$ during RL training.

## 6  DISCUSSION

We present a training-based approach to compress reasoning chains: Breadcrumbs Reasoning. Our empirical results demonstrate that indeed there is significant room for compression in reasoning chains, and not all information or tokens are equally important for downstream reasoning and even task completion. Our approach shows effective compression while retaining much of the reasoning performance. In contrast, training-free methods are significantly less effective.

We propose a joint RL–distillation training scheme, which provides an efficient way to teach a model to reason while also learning to compress. Training is necessary in any case to develop a policy that can successfully solve a reasoning task. Our approach integrates this requirement into a more effective procedure.

Although at a fixed generation length our method shows a small performance loss, we demonstrate that when generating many more tokens under the same memory budget, performance often surpasses that of a non-compressing teacher. This is enabled by effective test-time scaling: when

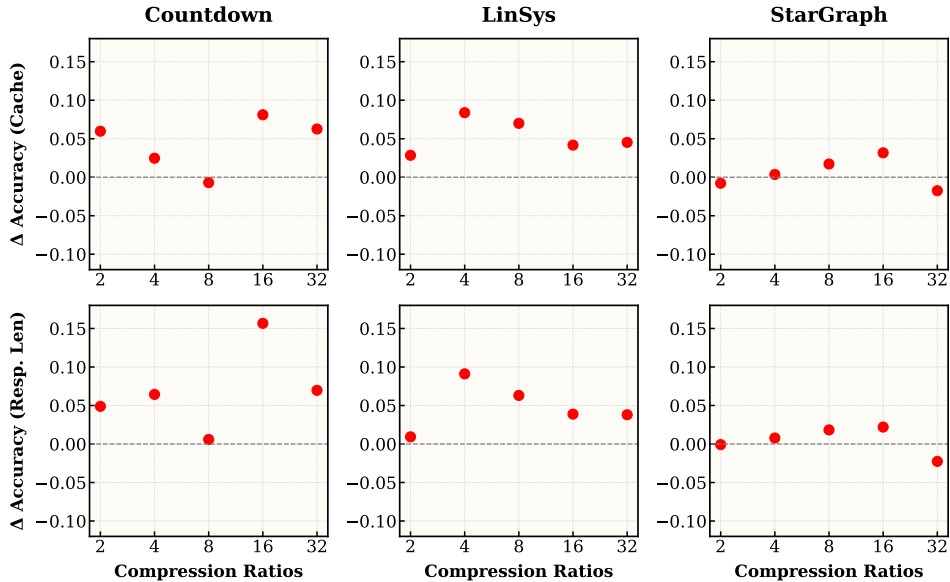

Figure 5: **Joint RL-distillation vs. Two-step Training on Qwen.** We compare our joint approach to the standard two-step process, where compression policies are trained on trajectories from a final $\pi_{\text{RL}}$ checkpoint. Each point shows the accuracy difference (Joint − Two-step) at a given compression ratio. The top row fixes cache size; the bottom fixes response length. Positive values favor joint training, which outperforms or matches in 26/30 settings, showing BR can learn compression online during RL without separate distillation data.

matching the KV-cache size, we are able to generate significantly more tokens. To some extent, these results show that we trade memory for time. Such trade-offs are common in efficiency-oriented methods — for example, speculative decoding (Leviathan et al., 2023; Chen et al., 2023) reduces latency but requires more memory, since multiple models must be loaded simultaneously.

There are several directions for future work, entailed from several areas where our approach can be improved. Foremost, while Multi Ratio Breadcrumbs Reasoning supports various compression rates, it lacks dynamic adaptivity. It does not automatically select the most appropriate ratio, and once a ratio is chosen, it remains fixed for the duration of the generation rather than adjusting dynamically across the sequence. This is an important direction for future work, and one that is relatively understudied in the compression literature. Our work also charts the direction for future work to improve test-time scaling and compression tradeoffs. Ideally, one can compress aggressively without needing to increase the number of reasoning steps. Our work exposes this tradeoff in reasoning models, and outlines the methodology to analyze it. Moreover, it would be interesting to explore how well our method combines with the orthogonal CoT shortening approach (Aggarwal & Welleck, 2025; Kang et al., 2025; Ma et al., 2025; Shen et al., 2025; Yan et al., 2025; Munkhbat et al., 2025; Xia et al., 2025). Methods in this direction train models to output shorter reasoning traces, but our method can still improve them by saving a significant fraction of memory. Even more, it is important to clarify that these methods do not claim that long reasoning chains are not necessary in general. In fact, the opposite has been theoretically proved by Merrill & Sabharwal (2023): as long as the Transformer architecture is used, long reasoning chains are still required as tasks get more complex. Finally, as the space of accessible benchmarking scenarios in reasoning research develops, it is important to understand the behavior of our approach across a broader set of domains.

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

## A    IMPLEMENTATION DETAILS

We use VeRL (Sheng et al., 2024) as the backbone for our training code. We adapt kvpress (Devoto et al., 2025), a recent library for Transformers KV cache compression for inference (e.g., for evaluation). We decouple RL and distillation during our experiments, so that the teacher data is the same for all BR policies and results can be compared more fairly. Otherwise, different experimental results may depend on the different quality of $\pi_{\mathrm{RL}}$ training, rather than on the methods being tested. We save all $\pi_{\mathrm{RL}}$ sampled trajectories to decouple training, and the top-100 logits for each time step, because saving all logits is extremely storage-intensive. In all training setups, this leads to consistently distilling more than 90% of the probability mass for each token, although it frequently reaches even higher levels throughout training.

**Overview of Beacon Learning**   Before training, we add to the vocabulary of the model a new token, the beacon. In the embedding matrix of the model, the beacon token is initialized as the average of all other embeddings. The model learns to compress normal tokens' key/value activations because, thanks to the attention mask in Figure 2, otherwise those activations' information would not be accessible from future tokens.

## B    TASK DETAILS

*Countdown*: The task is to combine 3–4 numbers using arithmetic operations to reach a target value. All target values are less than or equal to 100.

---

**Countdown Example Prompt**

Using the numbers `25, 10, 7, 3`, create an equation that equals `96`. You can use basic arithmetic operations (`+, -, *, /`) and each number can only be used once. Make sure to solve it by thinking step by step. Return the final answer in `<answer> </answer>` tags, for example `<answer> (1 + 2) / 3 </answer>`.

---

*StarGraph*: We use star graphs (Bachmann & Nagarajan, 2024) with branches of length 5 and up to 25 branches per graph. When creating the dataset, the number of branches for each graph is sampled uniformly from the range $[2, 25]$. We empirically found that this variable complexity helps the reinforcement learning policy, $\pi_{\mathrm{RL}}$, by allowing it to gradually learn to solve more complex graphs.

---

**StarGraph Example Prompt**

You are given a star graph with the following nodes: `34, 72, [...], 304`.

The graph has the following directed edges:
```
34 -> 72
[...]
```

Find the path from the center node `34` to the target node `304`.

Think step by step about the graph structure and trace the path from the center node to the target node.
Return your answer as a list of nodes representing the path from center to target.
Return the final answer in `<answer> </answer>` tags, for example `<answer>[1, 3, 7, 12]</answer>`.

---

*LinSys*: We generate systems of linear equations that have a single, unique solution. To ensure an appropriate level of difficulty for each model family, we used two distinct configurations:

- For Qwen models: We generated systems of 4 equations in 4 variables. Each equation has at most two non-zero coefficients, and the maximum absolute value for any coefficient is 20.

| PPO Hyperparameter | Value |
|---|---|
| Actor Learning rate | $1 \times 10^{-6}$ |
| Critic Learning rate | $1 \times 10^{-5}$ |
| Clip ratio ($\epsilon$) | 0.2 |
| Number of epochs | 1 |
| Mini-batch size | 256 |
| Discount factor ($\gamma$) | 1.0 |
| GAE parameter ($\lambda$) | 1.0 |
| Entropy coefficient | 0.001 |
| Value loss coefficient | 0.5 |
| Clip range value | 0.5 |
| Max gradient norm | 1.0 |

| AdamW Hyperparameter | Value |
|---|---|
| Weight decay | 0.01 |
| $\beta_1$ | 0.9 |
| $\beta_2$ | 0.999 |
| Epsilon ($\epsilon$) | $1 \times 10^{-8}$ |

Table 4: Hyperparameters for PPO (left) and AdamW (right) for $\pi_{\mathrm{RL}}$.

| Hyperparameter | Value |
|---|---|
| Weight decay | 0.01 |
| $\beta_1$ | 0.9 |
| $\beta_2$ | 0.999 |
| Epsilon ($\epsilon$) | $1 \times 10^{-8}$ |

Table 5: Hyperparameters for AdamW optimizer for $\pi_{\mathrm{BR}}$.

– For Phi models: We found the 4x4 configuration to be too simple (over 40% accuracy before training). We therefore created a more challenging setup: systems of 3 equations in 3 variables, with no restrictions on the number of non-zero coefficients and a maximum absolute coefficient value of 20.

---

**LinSys Example Prompt**

Solve the following system of linear equations:
```
2*x1 - x2 + 3*x3 + 4*x4 = 10
-x1 + 4*x2 - 2*x3 + x4 = 5
3*x1 + x2 + x3 - x4 = 12
x1 + 2*x2 + 5*x3 + 2*x4 = 20
```
Find the values for `x1, x2, ..., x4`. Make sure to solve it by thinking step by step, and do not assume access to any external tools.
Return the final answer as a list of numbers in `<answer> </answer>` tags, for example `<answer>[1, -2, 3, 7]</answer>`.

---

## C  Training Details

### C.1  $\pi_{\mathrm{RL}}$ Training

Table 4 provides the hyperparameters for PPO Schulman et al. (2017) and AdamW (Loshchilov & Hutter, 2019). The critic has the same architecture and initial weights as $\pi_{\mathrm{RL}}$.

### C.2  $\pi_{\mathrm{BR}}$ Training

Table 5 provides the hyperparameters for the AdamW optimizer. We use a learning rate of $5 \times 10^{-6}$ for all tasks and models, except for the Countdown-Phi configuration, where we use a higher learning rate of $5 \times 10^{-5}$. This choice is based on empirical observations.

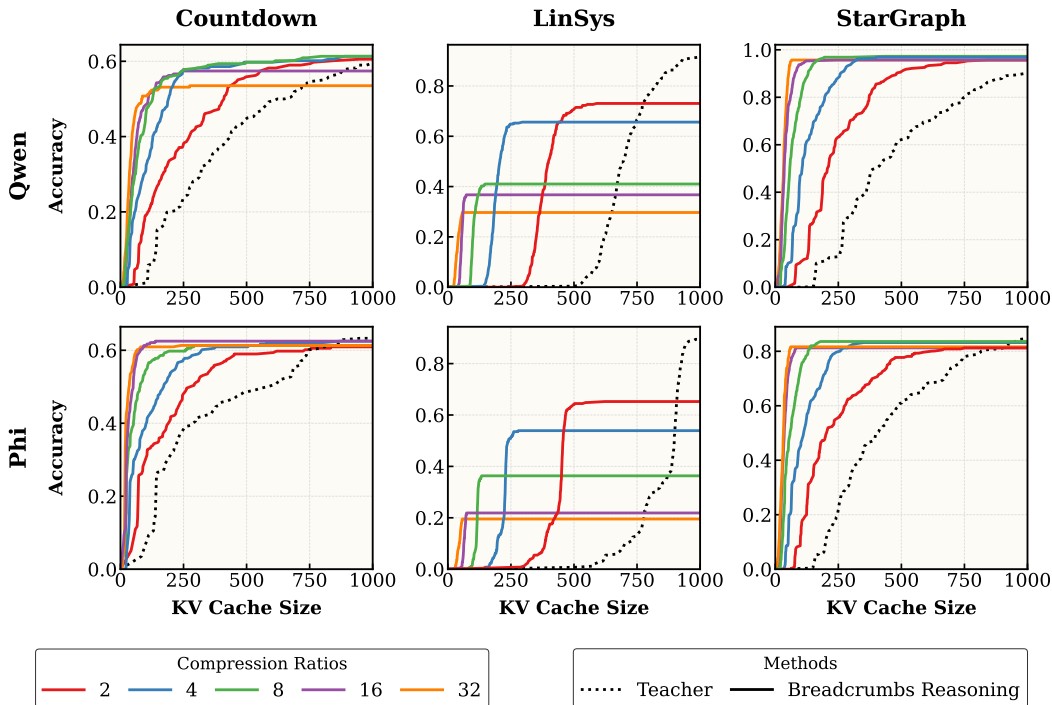

Figure 6: **Accuracy vs. KV Cache Size** Single Ratio Breadcrumbs Reasoning retains most of the teacher's performance while using significantly fewer KV cache entries, even outperforming the teacher when setting a fixed maximum KV cache size in Countdown and StarGraph.

## D   ANALYSIS

**LLM-as-a-Judge Pipeline.**   We analyze failures of Qwen on the LinSys task in two stages. First, for every incorrect generation we pair the model's reasoning trace with the ground truth and prompt a verifier LLM (Qwen3 30B A3B Thinking 2507) to identify the *first* erroneous step; the resulting snippet (e.g., "computed $100 - 3100$ as $-2999$") is stored. Second, we pass each extracted snippet to a classification judge that sees only the mistake text plus a fixed taxonomy (Arithmetic, Equation Misreading & Fabrication, Algebraic Manipulation, Output, Other), that we identify by manual inspection of the mistakes (and add Other to allow for cases that we do not explicitly categorize). Using best-of-$n$ sampling ($n=3$), the judge assigns the most specific category.

## E   ADDITIONAL RESULTS

We report in Table 6 the accuracy area under the curve (AUAC) metric with varying the maximum cache size up to 1,000.

Figure 6 shows the Pareto curves for Single Ratio Breadcrumbs Reasoning.

Figure 7 illustrates the increase in accuracy of BR when increasing the generation length from 1,000 to 1,000 $\times c$ (i.e., KV cache size of 1,000 entries).

## F   LLM USAGE

LLMs were used in the process of writing this paper to assist in creating tables and figures.

| | **QWEN** | | | | | | | | | | | | | | | | | |
| | **COUNTDOWN** | | | | | | **LINSYS** | | | | | | **STARGRAPH** | | | | | |
| COMPR. RATIO | 1 | 2 | 4 | 8 | 16 | 32 | 1 | 2 | 4 | 8 | 16 | 32 | 1 | 2 | 4 | 8 | 16 | 32 |
| NO COMPRESSION | | | | | | | | | | | | | | | | | | |
| TEACHER | 0.377 | - | - | - | - | - | 0.276 | - | - | - | - | - | 0.513 | - | - | - | - | - |
| BREADCRUMBS REASONING | | | | | | | | | | | | | | | | | | |
| SR BR (OURS) | - | **0.465** | **0.533** | 0.555 | 0.539 | 0.513 | - | **0.451** | **0.532** | 0.368 | 0.347 | 0.286 | - | **0.726** | **0.845** | 0.906 | 0.918 | 0.926 |
| MR BR (OURS) | - | 0.446 | 0.515 | **0.576** | **0.576** | **0.560** | - | 0.439 | 0.507 | **0.424** | **0.391** | **0.316** | - | 0.713 | 0.839 | **0.912** | **0.921** | **0.935** |
| TRAINING-FREE COMPRESSION | | | | | | | | | | | | | | | | | | |
| PYRAMIDKV | - | 0.375 | 0.201 | 0.111 | 0.077 | 0.011 | - | 0.000 | 0.000 | 0.000 | 0.000 | 0.000 | - | 0.487 | 0.459 | 0.491 | 0.463 | 0.526 |
| SNAPKV | - | 0.376 | 0.210 | 0.138 | 0.100 | 0.079 | - | 0.003 | 0.000 | 0.000 | 0.000 | 0.000 | - | 0.534 | 0.478 | 0.485 | 0.452 | 0.489 |
| TOVA | - | 0.447 | 0.272 | 0.167 | 0.183 | 0.199 | - | 0.000 | 0.000 | 0.000 | 0.000 | 0.000 | - | 0.526 | 0.417 | 0.424 | 0.414 | 0.423 |
| STREAMINGLLM | - | 0.007 | 0.015 | 0.024 | 0.049 | 0.090 | - | 0.000 | 0.000 | 0.000 | 0.000 | 0.000 | - | 0.038 | 0.032 | 0.023 | 0.041 | 0.105 |
| | **PHI** | | | | | | | | | | | | | | | | | |
| | **COUNTDOWN** | | | | | | **LINSYS** | | | | | | **STARGRAPH** | | | | | |
| COMPR. RATIO | 1 | 2 | 4 | 8 | 16 | 32 | 1 | 2 | 4 | 8 | 16 | 32 | 1 | 2 | 4 | 8 | 16 | 32 |
| NO COMPRESSION | | | | | | | | | | | | | | | | | | |
| TEACHER | 0.433 | - | - | - | - | - | 0.142 | - | - | - | - | - | 0.498 | - | - | - | - | - |
| BREADCRUMBS REASONING | | | | | | | | | | | | | | | | | | |
| SR BR (OURS) | - | **0.506** | **0.557** | 0.581 | **0.604** | 0.597 | - | 0.373 | 0.421 | 0.322 | 0.205 | 0.187 | - | **0.636** | **0.739** | **0.786** | **0.785** | **0.792** |
| MR BR (OURS) | - | 0.480 | 0.526 | **0.582** | 0.568 | **0.609** | - | 0.375 | **0.454** | **0.408** | **0.278** | **0.258** | - | 0.631 | 0.710 | 0.765 | 0.732 | 0.738 |
| TRAINING-FREE COMPRESSION | | | | | | | | | | | | | | | | | | |
| PYRAMIDKV | - | 0.417 | 0.225 | 0.180 | 0.019 | 0.000 | - | 0.011 | 0.000 | 0.000 | 0.000 | 0.000 | - | 0.503 | 0.366 | 0.292 | 0.327 | 0.358 |
| SNAPKV | - | 0.406 | 0.222 | 0.149 | 0.062 | 0.012 | - | 0.006 | 0.000 | 0.000 | 0.000 | 0.000 | - | 0.516 | 0.424 | 0.329 | 0.331 | 0.304 |
| TOVA | - | 0.196 | 0.064 | 0.050 | 0.046 | 0.094 | - | 0.000 | 0.000 | 0.000 | 0.000 | 0.000 | - | 0.621 | 0.504 | 0.433 | 0.436 | 0.378 |
| STREAMINGLLM | - | 0.014 | 0.018 | 0.060 | 0.150 | 0.300 | - | 0.000 | 0.000 | 0.000 | 0.000 | 0.000 | - | 0.137 | 0.017 | 0.029 | 0.089 | 0.097 |

Table 6: **Model performance (AUAC) on long-context reasoning tasks**. We report AUAC given a maximum cache size of 1,000 entries. Higher is better.

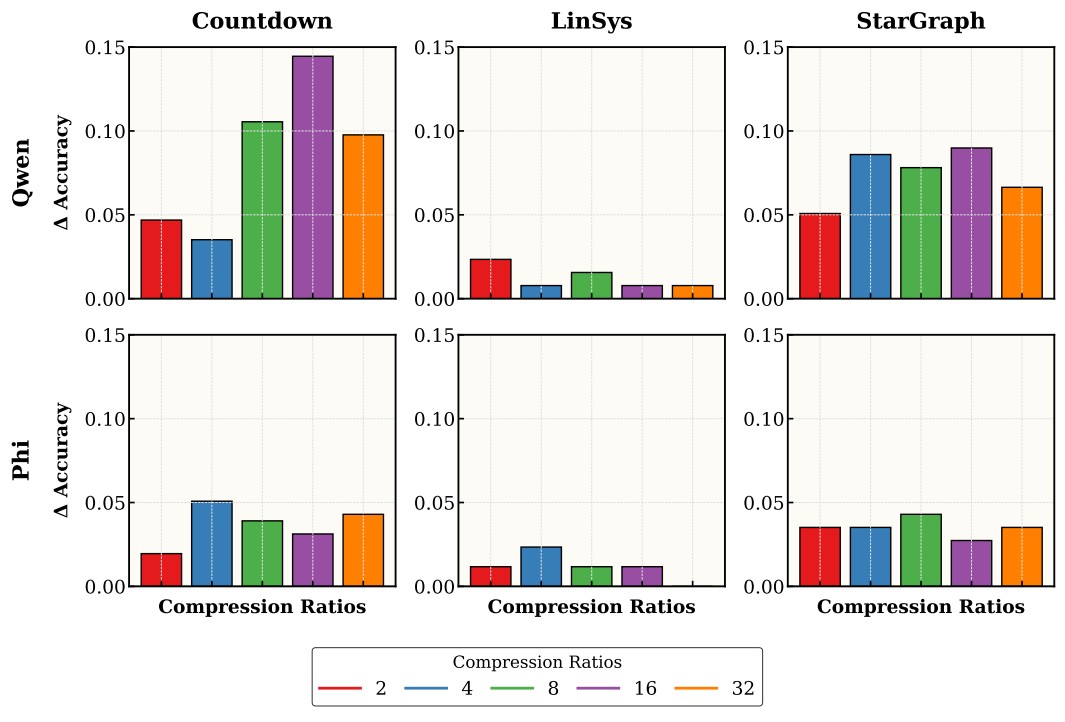

(a) Single Ratio Breadcrumb Reasoning

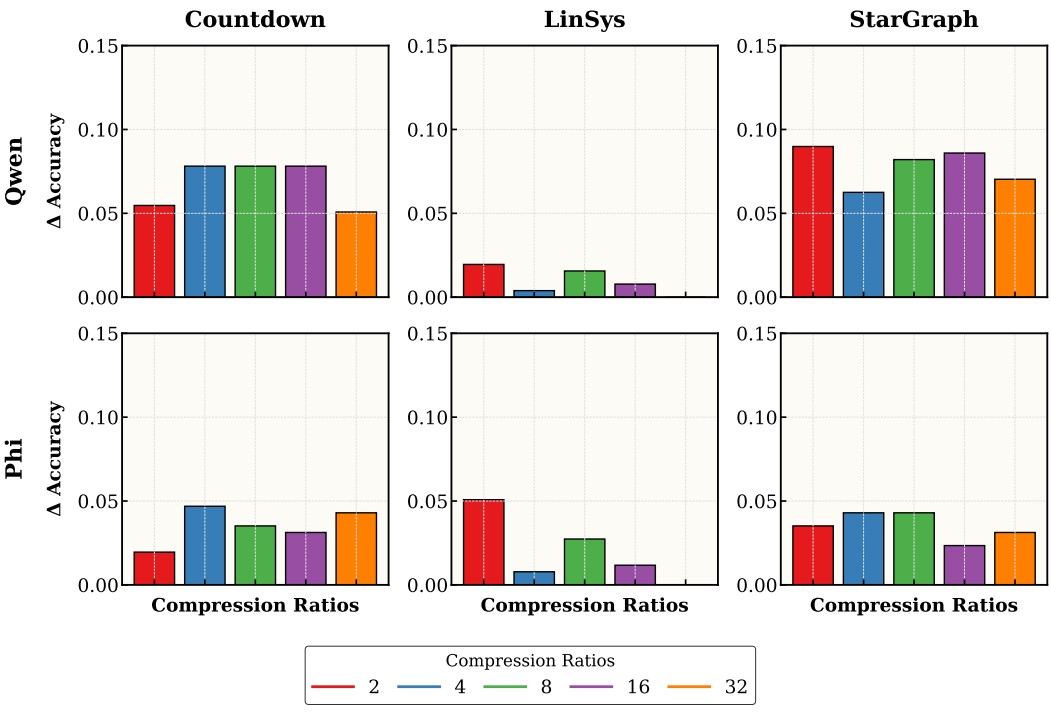

(b) Multi Ratio Breadcrumbs Reasoning

Figure 7: **Performance increase with extended generation.** Breadcrumbs Reasoning improves up to 14.5% with Qwen and 5.1% with Phi beyond the 1,000 token training length.

