# OpenReview forum: "Breadcrumbs Reasoning: Memory-Efficient Reasoning with Compression Beacons"
_ICLR.cc/2026/Conference — Submitted to ICLR 2026_

### Official Review · Reviewer_868j · 2025-10-31

**Soundness:** 2
**Presentation:** 3
**Contribution:** 2
**Rating:** 4
**Confidence:** 4

**Summary:**

This paper proposes a breadcrumb compression reasoning approach.  During long-context reasoning generation, a beacon token is inserted every c tokens. This beacon token learns to compress the key-value cache of the previous c tokens into a single entry. Then, the key-value cache of these c tokens is directly removed, retaining only the beacon token. This approach reduces the KV cache usage to 1/c of the original. The authors train this capability into the model using a joint RL+distillation training process. Compared to training-free cache pruning methods, this method delivers superior memory-accuracy Pareto frontiers on three inference tasks.

**Strengths:**

1. The proposed method is interesting and makes sense. Compressing the key-value cache of every 8 tokens in the context into a single beacon token is an intuitive approach.

2. The authors provide very detailed training pipelines in the paper.

3. The proposed method demonstrates strong results on the Countdown, Linsys, and Stargraph tasks.

**Weaknesses:**

1. Although the proposed method demonstrates strong results on Countdown, Linsys, and Stargraph, as a method specifically designed to optimize long-context reasoning, does it also perform well on mainstream mathematical reasoning tasks (such as MATH500 and AIME)?

2. The method demonstrates its effectiveness on the instruction models Qwen-2.5-1.5B-Instruct and Phi-4-Mini-Instruct. However, can it be directly applied to the long-context reasoning model (deepseek-distil-qwen-1.5b)? Or does it require an instruction model or base
model as a starting point for training?

3. Some important kv-cache pruning baselines, such as SnapKV and PyramidKV, are missing from the experiment.

4. The Contributions, despite the interesting methods, are not so significant.

5. The organization and formatting of the papers can indeed be improved.

**Questions:**

Please see the issues raised in the Weaknesses session.

It is critical to address questions 1, 2, 3, and 4, please.

---

> ### Author Response · Authors · 2025-11-22
>
> > Although the proposed method demonstrates strong results on Countdown, Linsys, and Stargraph, as a method specifically designed to optimize long-context reasoning, does it also perform well on mainstream mathematical reasoning tasks (such as MATH500 and AIME)?
>
> The core of our approach is learning both to compress and to reason to solve new tasks. We selected our tasks because models initially have very low accuracy on them (< 5%), but without compression they can learn to solve them through RL. This setup allows us to demonstrate that our method can learn both reasoning and compression simultaneously.
>
> It is currently challenging, without access to large model checkpoints, to reproduce the same setup for standard mathematical reasoning tasks like MATH500 and AIME, and therefore this is beyond the scope of this work. Our choice of tasks is similar to other academic papers in this space [1, 2, 3, 4, 5]. We believe we provide evidence so frontier labs would be motivated to train models to learn mathematical reasoning tasks with compression by reproducing our setup on intermediate checkpoints, as we demonstrate the same learning dynamics on tasks where models initially have little knowledge.
>
> > The method demonstrates its effectiveness on the instruction models Qwen-2.5-1.5B-Instruct and Phi-4-Mini-Instruct. However, can it be directly applied to the long-context reasoning model (deepseek-distil-qwen-1.5b)? Or does it require an instruction model or base model as a starting point for training?
>
> We focus on teaching models both how to reason and how to compress simultaneously. If LLMs can already reason well (as with deepseek-distil-qwen-1.5b), the task would be easier as the RL component could potentially be skipped and only KL distillation could be used to teach compression. However, we chose to focus on the harder problem of learning both reasoning and compression together, and therefore have not yet tested this on models that already have strong reasoning capabilities.
>
> > Some important kv-cache pruning baselines, such as SnapKV and PyramidKV, are missing from the experiment.
>
> Thank you for the suggestion. We have now integrated SnapKV and PyramidKV into all our experiments. We find that they actually perform worse than TOVA in most settings. We believe the reason is that these methods are designed to compress prefilling tokens and, as we find, do not generalize well to generation settings. Repeatedly applying them during generation does not work well. We have included complete results and details about how we tested these methods in the updated paper.
>
> > The organization and formatting of the papers can indeed be improved.
>
> We have improved the organization of the paper and the presentation of results (e.g., Table 1 and Table 2). It would be helpful to know if there is any part of the paper that was believed to be problematic.
>
> [1] The pitfalls of next-token prediction
>
> [2] The Belief State Transformer
>
> [3] Stream of Search (SoS): Learning to Search in Language
>
> [4] Cognitive Behaviors that Enable Self-Improving Reasoners, or, Four Habits of Highly Effective STaRs
>
> [5] Learning Adaptive Parallel Reasoning with Language Models

---

### Official Review · Reviewer_QxJL · 2025-10-31

**Soundness:** 3
**Presentation:** 2
**Contribution:** 3
**Rating:** 4
**Confidence:** 3

**Summary:**

Breadcrumbs Reasoning proposes a memory-efficient framework for long-context reasoning by periodically compressing the Transformer key–value cache with a learned “beacon” token. The model summarizes recent reasoning segments into compact embeddings and evicts redundant cache entries, reducing memory usage during generation. Trained via a joint reinforcement learning and distillation process, it learns to reason and compress simultaneously with minimal overhead. Evaluated on Countdown, LinSys, and StarGraph using Qwen2.5-1.5B and Phi-4-Mini, the method achieves a superior memory–accuracy trade-off, retaining 67-94% of baseline performance with up to 32× less memory and even surpassing the uncompressed teacher under the same cache budget, demonstrating that reasoning traces can be effectively compressed for efficient test-time scaling

**Strengths:**

1. The idea of periodically compressing previous Chain-of-Thought tokens through learned beacon representations is interesting and conceptually appealing, offering a new perspective on memory-efficient reasoning.
2. The method demonstrates strong flexibility, as it can be applied with different compression ratios, allowing users to balance accuracy and memory efficiency based on deployment constraints.

**Weaknesses:**

1. Some performance degradation is observed across tasks, even at moderate compression ratios (e.g., 2×), suggesting that the compression process may occasionally discard useful reasoning context.
2. The methodology section could be clarified further, particularly how the beacon token is constructed, integrated, and optimized during training.
3. It would be valuable to include additional baselines, such as Controlling How Long A Reasoning Model Thinks With Reinforcement Learning [1], to better position the method within the broader literature on reasoning efficiency.
3. The paper does not report inference speed or latency improvements, which are essential for evaluating the practical benefits of the proposed compression scheme.

[1] L1: Controlling How Long A Reasoning Model Thinks With Reinforcement Learning

**Questions:**

Please see the weaknesses.

---

> ### Author Response · Authors · 2025-11-22
>
> We thank the reviewer for their comments and questions.
>
> > Some performance degradation is observed across tasks, even at moderate compression ratios (e.g., 2×), suggesting that the compression process may occasionally discard useful reasoning context.
>
> While we acknowledge that some performance drops occur in LinSys, we would like to emphasize that baseline methods achieve virtually 0% accuracy on this task, suggesting it is in general a hard problem for compression. Additionally, we conducted a failure mode analysis of our method on this task and observed that the model is not forgetting previous steps, but rather making more arithmetic errors.
>
> > The methodology section could be clarified further, particularly how the beacon token is constructed, integrated, and optimized during training.
>
> Thank you for the feedback. We have updated our submission to clarify that the beacon token is added to the embedding matrix of the model (initialized as the average of all other tokens). The embedding of this token is then trained like any other learned parameter of the model: by aligning the next-token prediction logits (of normal tokens, not beacons) to the surrogate policy.
>
>
> > It would be valuable to include additional baselines, such as Controlling How Long A Reasoning Model Thinks With Reinforcement Learning [1], to better position the method within the broader literature on reasoning efficiency.
>
> We thank the reviewer for the reference, which we have included in our updated paper. As we mentioned in the Related Work and Discussion sections of our original submission, teaching a model to generate shorter reasoning chains is an orthogonal direction to learning how to summarize previous portions of the generation. First, these approaches could be combined to achieve even higher efficiency. Second, as demonstrated by [2], as long as the Transformer architecture is used, long reasoning chains are still required to solve increasingly complex tasks.
>
> > The paper does not report inference speed or latency improvements, which are essential for evaluating the practical benefits of the proposed compression scheme.
>
> Our method is a memory-efficiency method. As such, our primary goal is to minimize the memory consumption of the model, while accepting potential increases in latency. In fact, our method makes more forward passes at generation time (because of the additional compression tokens), and can also generate many more tokens **given the same memory budget** as the teacher (32× more tokens for a ratio of 32×). Therefore, some increase in latency is expected. However, we find that (1) latency is comparable even when allowing the model to generate as many tokens as needed to fill 1000 KV cache entries (e.g., 32,000 with a compression ratio of 32, versus only 1,000 for the uncompressed teacher), and (2) it is overall faster than all other training-free baselines. Thesre results are visualized in Table 3.
>
> [2] The Expressive Power of Transformers with Chain of Thought

---

### Official Review · Reviewer_iHta · 2025-10-31

**Soundness:** 3
**Presentation:** 3
**Contribution:** 2
**Rating:** 4
**Confidence:** 4

**Summary:**

This paper introduces "Breadcrumbs Reasoning," a method to mitigate the high memory cost of the Transformer KV cache during long-form reasoning. The model is trained to periodically compress its generation KV cache into a single "beacon" token and then evict the compressed entries. The core contribution is a joint RL-distillation framework where trajectories from a standard RL-trained "teacher" policy are used to simultaneously distill a "student" policy that learns both to reason and to compress. This joint training approach minimizes additional overhead. Experiments on Qwen2.5-1.5B and Phi-4-Mini models show this method achieves a superior memory-accuracy Pareto frontier. It can match or exceed the teacher's accuracy within a fixed memory budget by enabling more reasoning steps , and it significantly outperforms training-free compression baselines like TOVA and StreamingLLM.

**Strengths:**

Efficient Training Strategy: The joint RL-distillation framework is a key strength. It cleverly piggybacks on the standard RL training process, using its trajectories to simultaneously teach compression. This avoids a complex and data-intensive two-stage training pipeline, and the paper provides a clear validation (Fig. 5) that this joint approach is as effective, if not more so, than a sequential one.

Strong Empirical Results: The method demonstrates a clear and significant advantage over training-free baselines (TOVA, StreamingLLM), which seem incapable of handling complex reasoning tasks. This provides strong evidence that a learned, task-aware compression scheme is necessary for this domain.

Effective Test-Time Scaling: The paper clearly demonstrates the practical benefit of the method. By compressing the cache, the model can afford to generate many more reasoning tokens within the same memory budget, often leading to higher final accuracy than the uncompressed teacher (as seen in Table 1 and Figure 3). This is a valuable practical trade-off.

**Weaknesses:**

Task-Specific Training: The proposed method requires running the entire joint RL-distillation process for each specific reasoning task. This limits the "out-of-the-box" generality of a trained model.

Limited Model Scale: The experiments are conducted on relatively small models (1.5B and ~4B). While promising, it remains an open question how the training dynamics and compression effectiveness will scale to much larger.

Performance on Structured Reasoning: The method's performance improvement was least pronounced on the LinSys task, which the authors note requires more structured, step-by-step deduction. This may suggest that the "breadcrumb" compression is too lossy for tasks where all intermediate steps are critical, and that it performs best on tasks with more "trial-and-error" components.

**Questions:**

1. Models. The experiments are conducted on 1.5B and ~4B parameter models (Qwen2.5-1.5B and Phi-4-Mini). How is the performance and training stability of the joint RL-distillation framework expected to scale to significantly larger models (e.g., 7B, 14B)? Furthermore, have the authors considered applying this method to reasoning models like DeepSeek-Distill-Models Series?

2. Novelty. The paper positions itself relative to Activation Beacons [1] by noting simplifications (e.g., no chunk/sub-chunk distinction) . However, the core concept of a learned token summarizing a past window is also present in methods like Gist Tokens [2]. Could the authors further clarify the primary novelty? Is the main contribution the joint RL-distillation training strategy that adapts this compression specifically to reasoning tasks, rather than a fundamental difference in the compression mechanism itself?


3. Variable Compression. Have the authors experimented with training a single model to handle variable compression ratios? For instance, by stochastically sampling the ratio c during training, or providing the target ratio as a conditioning input to the model?


I would like to improve me scores if authors can solve my questions.


[1] Zhang, Peitian, et al. "Long context compression with activation beacon." arXiv preprint arXiv:2401.03462 (2024).

[2] Mu, Jesse, et al. "Learning to compress prompts with gist tokens." Advances in Neural Information Processing Systems 36 (2023): 19327-19352.

---

> ### Author Response · Authors · 2025-11-22
>
> We thank the reviewer for their comments and questions.
>
> > Models. The experiments are conducted on 1.5B and ~4B parameter models (Qwen2.5-1.5B and Phi-4-Mini). How is the performance and training stability of the joint RL-distillation framework expected to scale to significantly larger models (e.g., 7B, 14B)? Furthermore, have the authors considered applying this method to reasoning models like DeepSeek-Distill-Models Series?
>
> Larger models generally have greater capacity. Therefore, if anything, it should be more challenging to teach a model to reason and compress at a smaller scale than at a larger scale. Unfortunately, we are not able to train larger models due to compute constraints, but we have already demonstrated that 1.5B and 4B models follow the same trends. We have not yet applied this method to the DeepSeek-Distill-Models Series, but this would be an interesting direction for future work.
>
> > Novelty. The paper positions itself relative to Activation Beacons [1] by noting simplifications (e.g., no chunk/sub-chunk distinction) . However, the core concept of a learned token summarizing a past window is also present in methods like Gist Tokens [2]. Could the authors further clarify the primary novelty? Is the main contribution the joint RL-distillation training strategy that adapts this compression specifically to reasoning tasks, rather than a fundamental difference in the compression mechanism itself?
>
> We thank the reviewer for referring us to Gist tokens, which we have now included in our citations. This work is similar to Activation Beacons. However, Gist tokens apply compression only once, after observing the full prompt, while our compression is repeated periodically every compression ratio tokens. In fact, Gist tokens is designed for prefilling compression, while our method compresses during generation time. This difference is substantial and is likely one of the reasons why prefilling-oriented training-free baselines (PyramidKV, SnapKV) perform worse than generation-oriented training-free baselines (TOVA).
>
> > Variable Compression. Have the authors experimented with training a single model to handle variable compression ratios? For instance, by stochastically sampling the ratio c during training, or providing the target ratio as a conditioning input to the model?
>
> We agree this is an interesting idea. In our original submission, we pointed this out as a promising future work direction. However, we have now implemented this approach and observed that it works similarly to single-ratio models, further strengthening our results. We have updated our paper with the Multi Ratio Breadcrumbs Reasoning variant, which we include in all tables.

---

> ### Comment · Reviewer_iHta · 2025-11-28
>
> I appreciate the authors implementing the suggested "Multi-Ratio" variant. It is encouraging to see that the method can handle variable compression ratios without performance degradation, which significantly improves the practical applicability of the work.
>
> The distinction drawn between prompt compression (Gist tokens) and generation compression (Breadcrumbs) clarifies the novelty of the proposal. I agree that training-free baselines focused on pre-filling often struggle with the dynamic nature of reasoning generation.
>
> I understand the hardware constraints regarding larger models. While the trends between 1.5B and 4B are consistent, I maintain that verifying this on larger, more capable reasoning models (e.g., 7B-14B range) would significantly strengthen the paper, as larger models may have different attention redundancy patterns.
>
> Overall, the rebuttal has addressed some of my concerns, particularly regarding flexibility. I will conisder to raise my score.

---

### Official Review · Reviewer_yRRw · 2025-11-06

**Soundness:** 3
**Presentation:** 3
**Contribution:** 3
**Rating:** 6
**Confidence:** 3

**Summary:**

This paper proposes Breadcrumbs Reasoning (BR), a learned KV cache compression method for long-context reasoning in LLMs. The approach periodically inserts special "beacon" tokens every c tokens that compress information from the preceding window, allowing those KV cache entries to be evicted. The key innovation is a joint RL-distillation training scheme that trains a compression-enabled policy πBR​ by distilling from an uncompressed teacher policy πRL​ during RL training, eliminating the need for separate pretraining on general data. Evaluated on Countdown, LinSys, and StarGraph tasks using Qwen2.5-1.5B and Phi-4-Mini, the method achieves 2-32× memory reduction while retaining 67-94% of teacher performance, and demonstrates superior test-time scaling compared to training-free baselines (TOVA, StreamingLLM).

**Strengths:**

1. Practical training methodology: The joint RL-distillation approach is elegant and efficient. By reusing πRL​ rollouts for distillation, it adds minimal overhead (~2-3× vs. ~5-10× for two-stage approaches). The ablation (Figure 5) convincingly shows it matches or outperforms late distillation.

2. Effective test-time scaling: The ability to generate 32,000 tokens (32× beyond training length) while maintaining coherence is impressive. This enables aggressive test-time compute scaling within fixed memory budgets - a practically important capability.

3. Strong empirical results on applicable tasks: On Countdown and StarGraph, the method achieves excellent memory-accuracy tradeoffs. For example, on Phi-Countdown, BR reaches 0.60 accuracy with only ~100 cache entries vs. 800+ for the teacher.

4. Thorough experimental design: The AUAC metric is appropriate for evaluating memory-accuracy tradeoffs. The comparison across multiple compression ratios (2-32×) provides good coverage of the design space.

5. Architectural simplicity: Removing the chunk/sub-chunk hierarchy and specialized attention from prior work (Zhang et al., 2025) is a genuine contribution that improves implementability.

**Weaknesses:**

1. Task-dependent brittleness:
The dramatic performance collapse on LinSys (91.8% → 28.9% for Qwen at c=32) is concerning and inadequately explained. The paper speculates this relates to "structured reasoning" but provides no analysis. This suggests the method may be fundamentally unsuitable for dense deductive reasoning where all steps are tightly coupled. Request: Provide attention analysis showing what information is lost during LinSys compression vs. Countdown. Is this a failure of the compression mechanism or the training procedure?
2. Fixed compression rate is a major limitation:
Training separate models for each compression ratio is impractical. More critically, uniform compression is suboptimal - some reasoning steps (e.g., discovering key constraints) warrant preservation while others (exploratory dead-ends) can be aggressively compressed. This limitation should be prominently acknowledged as it significantly impacts practical deployment.
3. Missing ablations:
- Sensitivity to beacon token initialization
- Effect of distillation coefficient (if using weighted RL + distillation loss)
- Performance with non-uniform compression (e.g., adaptive c based on heuristics)
4. Memory-time tradeoff underemphasized: While framed as enabling "test-time scaling," the method fundamentally trades memory for time. At c=32, you generate 32× more tokens to achieve similar accuracy. Total FLOPs and latency increase proportionally. The paper should more clearly discuss when this tradeoff is favorable (memory-constrained but not latency-sensitive scenarios).

**Questions:**

1. LinSys failure mode: Can you provide deeper analysis of why LinSys fails? Specifically:
- What information is being lost during compression? (attention weight analysis, probing classifiers)
- Does increasing beacon token dimension help?
- Is the issue that LinSys requires dense bidirectional reasoning that is incompatible with autoregressive compression?

2. Beacon content: What information do beacons actually encode? Can you:
- Visualize beacon embeddings (t-SNE, PCA)
- Measure mutual information between beacons and their compressed windows
- Show how beacon quality degrades with compression ratio

3. Comparison justification: Why were other learned compression methods (AutoCompressor, GIST) not compared? Are they incompatible with the reasoning setting?

---

> ### Author Response · Authors · 2025-11-22
>
> We thank the reviewer for their comments and questions.
>
> > LinSys failure mode: Can you provide deeper analysis of why LinSys fails? Specifically:
> What information is being lost during compression? (attention weight analysis, probing classifiers)
> Does increasing beacon token dimension help?
> Is the issue that LinSys requires dense bidirectional reasoning that is incompatible with autoregressive compression?
>
> We appreciate the question regarding the limitation of our method on LinSys. While we note that all baselines achieve virtually zero accuracy on this task (Table 1 and Table 2), it is important to understand why BR does not match the uncompressed teacher performance as in the other settings. We conducted an analysis of the rollouts generated by BR and found that with increased compression, the model makes progressively more arithmetic errors, while the number of fabrications and memorization errors (e.g., re-finding a variable that it has already identified) represents only a small fraction of the total errors. One potential factor is that compressing numbers might be harder, and also that our compression method might break some arithmetic circuits of the model, and that our training is not long enough to alleviate these issues. We leave finding remedies for this as an important direction for future work.
>
>
> > Beacon content: What information do beacons actually encode? Can you:
> Visualize beacon embeddings (t-SNE, PCA)
> Measure mutual information between beacons and their compressed windows
> Show how beacon quality degrades with compression ratio
>
> These are interesting questions that would provide valuable insights into the compression mechanism. A thorough analysis of beacon content (e.g., visualization, mutual information measurements) is beyond the scope of this work, which focuses on demonstrating the effectiveness of the overall approach. We have observed that performance degrades with increased compression (as shown in our LinSys results), which provides indirect evidence of compression quality. A detailed analysis of what information beacons encode would be a valuable direction for future work.
>
> > Comparison justification: Why were other learned compression methods (AutoCompressor, GIST) not compared? Are they incompatible with the reasoning setting?
>
> Both AutoCompressor and GIST focus on prefilling (i.e., prompt) compression, applying compression only once after observing the full prompt. In contrast, our method is designed to compress repeatedly during generation time, as generation continues. This fundamental difference makes these methods inapplicable to our setting, as they are not designed for the dynamic compression required during autoregressive generation. We have now included GIST in our citations and discuss this distinction in the Related Work section.

---

### Author Response · Authors · 2025-11-22
**Update to the Paper**

We thank the reviewers for taking the time to read our paper and provide helpful and insightful feedback. Based on their comments, we have extended our paper to address the following questions.

## Single Model for All Compression Ratios

> You are training one model per compression ratio and this is expensive. It would be cool if you could train a single model on all compression ratios and then decide at test time which ratio to use.

**TL;DR:** We implement this approach and find that it works even better than single-ratio training. Further demonstrating the effectiveness of our approach.

We agree that it would be much more compelling if the model could be trained once on all compression ratios and then the appropriate ratio could be chosen at test time for each input. We conducted additional experiments to train a model capable of this. We distinguish between two variants of our method: **Single Ratio Breadcrumbs Reasoning (SR BR)**, the version proposed in the original submission, and **Multi Ratio Breadcrumbs Reasoning (MR BR)**. We find that MR BR on average outperforms SR BR. Please refer to the updated paper for more details and complete results.

## Additional Baselines

> Can you add more baselines (SnapKV, PyramidKV)?

**TL;DR:** Done. SnapKV and PyramidKV perform worse than other baselines.

SnapKV and PyramidKV are recent SOTA prefill KV cache compression techniques. We did not initially test them because, unlike StreamingLLM and TOVA, they are not designed for compression during generation time and are applied only once at the beginning of generation. Nonetheless, these methods could potentially work if simply applied periodically every compression ratio tokens, mimicking our approach. However, we find that this is not the case—they actually perform worse than TOVA in most settings. We include complete results from these two additional baselines in all our tables.

## LinSys Underperformance Analysis

> Why does LinSys underperform compared to the other two tasks?

**TL;DR:** The compressing model does not forget more information, but tends to make more arithmetic mistakes.

We appreciate the question regarding the limitation of our method on LinSys. While we note that all baselines achieve virtually zero accuracy on this task (Table 1 and Table 2), it is important to understand why BR does not match the uncompressed teacher performance as in the other settings. We conducted an analysis of the rollouts generated by BR and found that with increased compression, the model makes progressively more arithmetic errors, while the number of fabrications and memorization errors (e.g., re-finding a variable that it has already identified) represents only a small fraction of the total errors.

## Additional Improvements

We have also addressed minor points, improved the organization of the paper (e.g., tables are now clearer), and included latency information.

---

### Author Response · Authors · 2025-12-02

Although the reviewers-authors discussion period was impacted by the OpenReview leakage and the subsequent decision by ICLR program chairs to disable author-reviewer discussion, we implemented significant updates that directly address the reviewers' major concerns. While three out of four reviewers did not have the opportunity to review these updates before the discussion closed, we wish to highlight the following critical points:

**1. Addressed Major Concerns with New Experiments**
We improved our experimental setup by incorporating suggestions directly from the reviews:
* **Multi-Ratio Compression:** We added an investigation initially proposed for future work: a single model capable of handling multiple compression ratios. We show this works and, on average, performs even better than single-ratio models. This addition clarifies and resolves major concerns from **iHta** and **yRRw**, strengthening our results.
* **Additional SOTA Baselines:** To improve soundness, we included additional SOTA baselines (SnapKV and PyramidKV) as recommended by reviewer **868j**. These baselines fail just like other KV cache removal methods, confirming the effectiveness of our method.
* **Failure Mode Analysis:** We added an analysis of failure modes with high compression on Linear System. We show that models retain memory but exhibit arithmetic mistakes. This clarifies concerns from **yRRw** and **QxJL** regarding potential forgetting, demonstrating that forgetting is not occurring.

**2. Positive Reception from Rebuttals**
Reviewer **iHta**, who was able to see our clarification, explicitly agreed on the novelty of our method (noting our rebuttal “clarifies the novelty of the proposal”) and stated that our new results “significantly improves the practical applicability of the work.” Notably, **iHta** mentioned they would have raised their score (changing their comment from “I will raise my score” to “I will consider to raise my score” shortly after, likely due to the program chairs’ decision to disable the edit button for reviews). We believe other reviewers would have similarly appreciated these updates as we directly implemented their suggestions.

**3. Consensus on Strengths**
Overall, all reviewers appreciated our contributions across three key areas:

* **On the training side:**
    * “The joint RL-distillation approach is elegant and efficient”
    * “The joint RL-distillation framework is a key strength. It cleverly piggybacks on the standard RL training process”
* **On the compression strategy:**
    * “The proposed method is interesting and makes sense. Compressing the key-value cache of every 8 tokens in the context into a single beacon token is an intuitive approach”
    * “The idea of periodically compressing previous Chain-of-Thought tokens through learned beacon representations is interesting and conceptually appealing, offering a new perspective on memory-efficient reasoning.”
* **On our results:**
    * “Strong Empirical Results”
    * “Strong empirical results on applicable tasks”
    * “The proposed method demonstrates strong results on the Countdown, Linsys, and Stargraph tasks”
    * “The ability to generate 32,000 tokens (32× beyond training length) while maintaining coherence is impressive”

---

### Meta-Review · Area_Chair_ELiz · 2026-01-06

**Summary:**

The final review score leans to the rejection side. Despite the rebuttal, the paper still has major issues not fully addressed, e.g. limitation on structured reasoning, limited interpretability of the compression mechanism, and restricted validation scope (tasks and model). Key limitations are acknowledged but largely deferred to future study. Overall, the contribution is solid but not mature for acceptance at this time.

**Reviewer Concerns:**

- The rebuttal is brief and did not fully address the raised points. For example, the weakness points from reviewer yRRw.
- The performance drop on LinSys is brought up by multiple reviewers, with the concern that this method may have a major issue when applied to problems whose intermediate steps are critical. The explanation in the rebuttal is that "this is due to arithmetic mistakes but there's no forgetting" I find it contradictory, as if enough information is retained it should have done the arithmetic correctly.
- The requests for evaluation on larger model, math task, and study of what beacons encode are deferred to future work.

**Reviewer Scores:**

The current score is 6444, iHta mentioned consideration of raising score, in that case the overall score would be borderline. I don't have strong evidence that the other reviewers will raise their score given the above mentioned reasons.

---

### Decision · Program_Chairs · 2026-01-26

Reject